# A Successful Bridge Therapy Combining Hypomethylating Agents with Venetoclax for Adult Patients with Newly Diagnosed or Relapsed/Refractory Acute Myeloid Leukemia

**DOI:** 10.3390/cancers15061666

**Published:** 2023-03-08

**Authors:** Su-Yeon Bang, Silvia Park, Daehun Kwag, Jong Hyuk Lee, Gi-June Min, Sung-Soo Park, Jae-Ho Yoon, Sung-Eun Lee, Byung-Sik Cho, Ki-Seong Eom, Yoo-Jin Kim, Seok Lee, Chang-Ki Min, Seok-Goo Cho, Jong Wook Lee, Hee-Je Kim

**Affiliations:** 1Department of Hematology, Catholic Hematology Hospital, Seoul St. Mary’s Hospital, College of Medicine, The Catholic University of Korea, Seoul 06591, Republic of Korea; 2Leukemia Research Institute, College of Medicine, The Catholic University of Korea, Seoul 06591, Republic of Korea

**Keywords:** acute myeloid leukemia, venetoclax, hypomethylating agents, allogeneic transplantation

## Abstract

**Simple Summary:**

The introduction of venetoclax (VEN) to combination regimens has dramatically changed the paradigm of treatment for acute myeloid leukemia (AML) patients deemed unfit for intensive chemotherapy. Several researchers have reported that this regimen was successfully used as a bridge to potentially curative allogeneic hematopoietic stem cell transplantation (allo-HCT) in AML patients; however, data about clinical outcomes after allo-HCT are still lacking. In this study, we evaluated the post-transplant outcomes of 50 patients who received VEN-hypomethylating agents (HMA) treatment, either as initial therapy for newly diagnosed AML (*n* = 10) or as a salvage regimen for relapsed/refractory (R/R) AML (*n* = 40). The probabilities of overall survival, relapse-free survival, cumulative incidence of relapse, and nonrelapse mortality at 1 year were 63.7%, 59.3%, 28.5%, and 12.2%, respectively. Even though 30% of our data were from a second allo-HCT, our results suggest that allo-HCT following VEN-HMA therapy is a safe and effective treatment option.

**Abstract:**

Recently, the combination of VEN-HMA has been shown to achieve durable responses in patients with both newly diagnosed (ND) and R/R-AML. We retrospectively evaluated the post-allo-HCT outcomes of 50 patients who received VEN-HMA therapy. In total, 10 were ND and 40 were R/R and, at the time of HCT, the median age was 53 years. In the ND- and R/R-AML groups, the percentage of patients who achieved CR/CRi or MLFS was 90% and 92.5%, respectively. In all, after a median follow-up of 13.7 months, the probabilities of overall survival (OS), relapse-free survival (RFS), cumulative incidence of relapse (CIR), and nonrelapse mortality (NRM) at 1 year were 63.7%, 59.3%, 28.5%, and 12.2%, respectively. In addition, the cumulative incidences of grade II–IV acute graft-versus-host disease (GVHD) and moderate–severe chronic GVHD at 1 year were 28.4% and 37.4%, respectively. In multivariate analysis, the factors associated with a statistically significant impact on OS were VEN-HMA cycle (*p* = 0.021), ELN risk group (*p* = 0.041), and the response to VEN-HMA therapy before allo-HCT (*p* = 0.003). Although 80% of our patients had R/R-AML and 30% underwent a second allo-HCT, our data still suggest that allo-HCT following VEN-HMA therapy is a safe and effective treatment option.

## 1. Introduction

The combination of anthracycline and cytarabine (3 + 7) intensive chemotherapy (IC) has been the mainstay of treatment for newly diagnosed (ND) acute myeloid leukemia (AML) patients for decades, with complete remission (CR) rates of 70–80% and long-term survival rates of 30–40% [1,2,3]. However, many AML patients are poor candidates for IC due to their age, poor performance status, or multiple comorbidities, and they have generally been offered lower-intensity treatment, such as low-dose Ara-C (LDAC) and hypomethylating agents (HMA), or just best supportive care [4].

Recently, the introduction of venetoclax (VEN) to combined regimens has deeply changed the paradigm of treatment for AML patients deemed medically unfit for IC. VEN-based combinations with LDAC or azacitidine (AZA) are currently approved as frontline therapy for this population, and they produced a CR or CR with incomplete count recovery (CRi) in up to 67% of patients in the pivotal trial that led to VEN approval [5,6]. Although the approval for use is currently confined to patients with ND-AML, mounting evidence indicates increasing off-label use of these combinations to treat refractory/relapsed (R/R) AML because of their efficacy and safety profile. According to a series of reports [7,8,9,10], VEN-HMA combinations have yielded response rates as high as 64% in R/R-AML patients [8], which appears to be noninferior to results with salvage IC [10], even though the treatment intensity of the VEN combination is much lower than that of conventional cytotoxic salvage regimens.

Although novel therapeutics have produced major improvements in the outcomes of AML patients, the long-term outcomes of elderly patients with AML or R/R-AML patients remain disappointing, and very few patients who do not undergo allogeneic hematopoietic stem cell transplantation (allo-HCT) are long-term survivors [11,12]. Given the fact that allo-HCT remains the sole curative option for the majority of this population, the high response rates achieved with VEN-HMA combinations could be expected to allow more patients to proceed to allo-HCT with curative intent [13].

Indeed, several reports describe the successful use of this regimen as a bridge to potentially curative allo-HCT in elderly patients with ND-AML or R/R-AML [4,7,10,13,14,15,16,17]; however, data about clinical outcomes after allo-HCT are still lacking. In this study, we evaluated the post-transplant outcomes of patients who received VEN-HMA as initial therapy for ND-AML or as a salvage regimen for R/R-AML.

## 2. Methods

### 2.1. Study Population and Data Selection

The cohort consisted of 181 patients with ND- or R/R-AML (age ≥ 18 years) who received VEN-HMA therapy at Seoul St. Mary’s Hospital between February 2020 and February 2022. Those who had not yet undergone transplantation (*n* = 67) and those who died before transplantation (*n* = 51) were excluded. Of the 63 patients who received HCT between June 2020 and February 2022, 10 ND- and 40 R/R-AML patients who received VEN-HMA treatment as their last therapy before allo-HCT were included, excluding 2 who received auto-HCT (Figure 1). Information about the patients and their disease status, transplant characteristics, and outcomes were retrospectively reviewed using data from the electronic medical record system. This study was approved by the Institutional Review Board and Ethics Committee of the Catholic Medical Center in South Korea (KC23RASI0081).

### 2.2. VEN-HMA Treatment and Response Assessment

For the VEN-HMA therapy regimen, the physician may opt to use either intravenous decitabine (DEC) at a dose of 20 mg/m^2^ on days 1–5 or subcutaneous AZA at a dose of 75 mg/m^2^ on days 1–7 in combination with VEN, which is administered at a dose of 100 mg on day 1, 200 mg on day 2, and 400 mg on days 3–28 of cycle 1 [5,6]. From the second cycle, VEN was started at 400 mg. Dose interactions between VEN and concomitant medications (including antifungal agents) were considered, and the VEN dose was adjusted as previously described [10].

The response to VEN-HMA before transplantation was defined according to consensus criteria as follows [18,19]: CR: bone marrow blasts <5%, absence of circulating blasts and blasts with Auer rods, absence of extramedullary disease, absolute neutrophil count ≥1.0 × 10^9^/L (1000/mL), platelet count ≥100 × 10^9^/L (100,000/mL); CRi: all criteria for CR met except for residual neutropenia (<1.0 × 10^9^/L (1000/mL)) or thrombocytopenia (<100 × 10^9^/L (100,000/mL)); morphologic leukemia-free state (MLFS): all criteria for CR met except for residual neutropenia (<1.0 × 10^9^/L (1000/mL)) and thrombocytopenia (<100 × 10^9^/L (100,000/mL)). Patients with >5% bone marrow myeloblasts were classified as having refractory disease. The classification of measurable residual disease (MRD) positivity was determined based on the presence of abnormalities in the following order: (1) <3 log reduction in *RUNX1-RUNX1T1* and *CBFB-MYH11*, (2) <4 log reduction in pretransplant *NPM1* level from diagnosis, and (3) pretransplant *WT1* levels ≥ 250 copies without *RUNX1-RUNX1T1*, *CBFB-MYH11*, or *NPM1* mutations [20].

### 2.3. Transplant Procedure

The physician used their discretion to select the conditioning regimens, taking into account various factors, such as recipient age, comorbidities, donor type, and performance status of the patient at the time of HCT. At our institution, the mainstay regimens for young patients undergoing HCT are myeloablative conditioning (MAC) protocols, which are chosen based on whether the patient has a matched sibling donor (MSD), unrelated donor (UD), or haploidentical donor (HID). For MSD or UD transplantation, the Bu-Cy regimen is used, consisting of busulfan at a dose of 3.2 mg/kg/day for 4 days from D-7 to D-4 and cyclophosphamide at a dose of 60 mg/kg/day for 2 days from D-3 to D-2. For HID transplantation, the Flu-Bu2-TBI800 regimen is used, which includes total body irradiation (TBI) at a dose of 400 cGy/day for 2 days on D-9 to D-8, fludarabine at a dose of 30 mg/m^2^/day for 5 days from D-7 to D-3, and busulfan at a dose of 3.2 mg/kg/day for 2 days from D-6 to D-5. The Cy-TBI regimen, which involves the administration of cyclophosphamide at a dose of 60 mg/kg/day for 2 days from D-7 to D-6 and TBI at a dose of 330 cGy/day for 4 days from D-5 to D-2, as well as the Flu-Ara-C-TBI regimen, which includes fludarabine at a dose of 30 mg/m^2^/day for 5 days from D-9 to D-5, cytarabine at a dose of 3 g/m^2^/day for 3 days from D-9 to D-7, and TBI at a dose of 400 cGy/day for 3 days on D-4 to D-2, were also utilized. However, the Flu-Ara-C-TBI regimen was only used in patients who underwent cord blood transplantation (CBT). Flu-Bu2-TBI400 (fludarabine 30 mg/m^2^/day for 5 days from D-6 to D-2, busulfan 3.2 mg/kg/day for 2 days from D-5 to D-4, and TBI 400 cGy/day for 1 day on D-1) was the primary reduced-intensity conditioning (RIC) protocol in our institution, while the prevention of graft-versus-host disease (GVHD) varied based on the human leukocyte antigen-matching status and type of transplant donor. GVHD prevention involved the use of either cyclosporine in the case of transplantation from an MSD or tacrolimus for transplantation from other donors, along with methotrexate. Patients who underwent umbilical CBT were administered mycophenolate mofetil thrice a day at a dosage of 3 g. Patients who had a UD or HID received a dosage of 1.25 mg/kg of anti-thymocyte globulin (ATG) once daily for 2 days (D-3 to D-2) and 4 days (D-4 to D-1), respectively, whereas it was not a standard practice to administer ATG to patients who had an MSD [21,22].

### 2.4. CMV Prophylaxis

CMV prophylaxis was indicated for patients who had positive CMV serologic status with an undetectable level of CMV DNAemia in whole blood before allo-HCT. Those patients received letermovir (LTM; 480 mg tablet once daily between day 0 and day +28 after allo-HCT and continued until day +100 if no adverse events occurred during the observation period). If LTM was co-administered with cyclosporine, the LTM dosage was decreased to 240 mg once daily [23].

### 2.5. Definition and Assessment of Outcomes

The overall survival (OS), relapse-free survival (RFS), cumulative incidence of relapse (CIR), and nonrelapse mortality (NRM) were calculated from the date of allo-HCT. OS was defined as death from any cause. For RFS, relapse and death, whichever occurred first, were considered uncensored events. For CIR, relapse was assessed as uncensored events, and death in CR was considered a competing cause of failure. NRM was defined as death with relapse as a competing risk. Acute graft-versus-host disease (GVHD) (aGVHD) was graded according to consensus grading criteria [24] and chronic GVHD (cGVHD) according to the National Institutes of Health (NIH) criteria [25].

### 2.6. Statistical Analysis

Descriptive statistics were utilized to demonstrate the frequencies and distribution of baseline characteristics. To determine differences between groups, categorical variables were analyzed using the Chi-square test or Fisher exact test, while continuous variables were assessed using a 2-sample *t* test or Mann–Whitney U-test. OS and RFS curves were plotted using the Kaplan–Meier method and analyzed with the log-rank test. The association between OS, RFS, and clinical and laboratory characteristics was evaluated using the Cox proportional hazard regression model. CIR, NRM, and cumulative incidence of aGVHD and cGVHD were estimated in a competing risk framework using the cumulative incidence of competing events, Gray test for univariate analysis, and Fine–Gray proportional hazard regression for multivariate analysis. Variables that yielded a *p*-value < 0.1, as determined by univariate analysis, were considered eligible for inclusion in the multivariate analysis. Additionally, the hematopoietic cell transplantation–comorbidity index (HCT-CI) and the response to VEN-HMA therapy before allo-HCT were included in the multivariate analysis as variables of interest, regardless of their *p*-values from univariate analysis. All statistics were conducted using IBM SPSS statistics version 25 and EZR software [26] v. 1.40.

## 3. Results

### 3.1. Patient’s Characteristics

The baseline characteristics of the patients and their allo-HCT are described in Table 1. Among the 50 patients who received VEN-HMA as their last therapy before undergoing allo-HCT, 10 (20.0%) and 40 (80.0%) patients had ND-AML and R/R-AML, respectively. Among the ND-AML patients (*n* = 10), the median age at diagnosis was 67 years and male sex was predominant (70.0%). According to the ELN 2022 risk classification [27], the favorable (FAV), intermediate (INT), and adverse (ADV) risk groups comprised 10.0%, 50.0%, and 40.0%, respectively, and most patients (70.0%) had INT-risk cytogenetics. Cytogenetics and genetic mutation tests required for ELN classification were performed in all patients as previously described [21,28,29,30,31]. All patients were treated with DEC as the agent combined with VEN, and patients received a median of four cycles (range, 3–7) of VEN-DEC prior to allo-HCT. Among the 40 R/R-AML patients, the median age at diagnosis was 49.5 years, and 15 patients (37.5%) had received prior allo-HCT before their treatment with VEN-HMA. Using the ELN 2022 criteria, the FAV, INT, and ADV risk groups comprised 12.5%, 40.0%, and 47.5% of patients, respectively. Only one patient received AZA as a partner HMA to VEN (2.5%), and the remaining patients received DEC (97.5%). The number of VEN-HMA cycles that patients received ranged from one to six, with a median of two cycles. When comparing the baseline characteristics between the ND- and R/R-AML groups, the ND-AML patients were significantly older than those with R/R-AML (67 vs. 49.5 years, *p* < 0.001).

### 3.2. Response to VEN-HMA before Allo-HCT and Transplant Characteristics

Prior to allo-HCT, the percentage of patients achieving CR/CRi or MLFS after VEN-HMA treatment was 90% (9/10) and 92.5% (37/40) in the ND- and R/R-AML groups, respectively (*p* = 0.794). Using the HCT-CI categories of low (0), intermediate (1 or 2), and high (≥3) risk, the high-risk group comprised 60.0% of the ND-AML group and 32.5% of the R/R-AML group (*p* = 0.283). In both the ND- and R/R-AML groups, UD was the most common type of donor, accounting for 50.0% in both settings. Next to UD transplantation, allo-HCT from HID was frequently performed in ND-AML patients (40.0%) and MSD transplantation was unusual in this group (10.0%). Among the R/R-AML patients, MSD and HID transplantations were equally common (20.0%) after UD transplantation, with CBT performed in four cases (10.0%), all but one of which were second transplants. Regarding conditioning intensities, MAC and RIC were used equally (50%) in ND-AML patients, whereas most R/R-AML patients (90.0%) received MAC (*p* = 0.010).

### 3.3. Post-Transplant Survival and Relapse Outcomes

The median duration of follow-up for all patients was 13.7 months (range, 11.0 to 16.4). The estimated median OS was not reached, and 1-year OS was 63.7% (95% CI 47.3% to 76.3%) for all patients. Among ND- and R/R-AML patients, post-transplant OS at 1 year was 77.1% (95% CI 34.5% to 93.9%) and 60.5% (95% CI 42.0% to 74.7%), respectively (Figure 2A). The median RFS was not reached for the entire patient cohort and 1-year RFS was estimated to be 59.3% (95% CI 43.6% to 72.1%) for all patients, 80.0% (95% CI 40.9% to 94.6%) for ND-AML, and 54.3% (95% CI 36.7% to 68.9%) for R/R-AML patients (Figure 2B). At 1 year after allo-HCT, CIR among the entire cohort was 28.5% (95% CI 16.0% to 42.4%), 10.0% (95% CI 4.0% to 37.6%) for ND-AML patients, and 33.0% (95% CI 18.0% to 48.7%) for R/R-AML patients (Figure 2C). The 1-year NRM was 12.2% (95% CI 4.9% to 23.0%) overall, 10.0% (95% CI 0.5% to 37.4%) for ND-AML patients, and 12.7% (95% CI 4.6% to 25.3%) for R/R-AML patients (Figure 2D, Table 2).

### 3.4. Post-Transplant Complications and Mortality

Overall, the cumulative incidence of aGVHD grade II–IV at 100 days was 28.4%, of which 8.4% was grade III–IV (Figure 3A,B). R/R-AML patients tended to have higher incidence of aGVHD grade II–IV than ND-AML patients (30.3% vs. 20.0%, *p* = 0.634). The 1-year cumulative incidence of cGVHD of NIH grade moderate to severe was 37.4%, with a 20.9% rate of severe-grade cGVHD in all patients (Figure 3C,D); those rates were 15.6% and 0% for ND-AML patients and 39.3% and 23.9% for R/R-AML patients, respectively (Table 2). A total of 7 ND-AML patients (70.0%) and 15 R/R-AML patients (37.5%) tested positive in at least one polymerase chain reaction assay for CMV DNA at a median of 71 days (range, 17 to 149) after allo-HCT, and 5 (50.0%) ND patients and 4 (10.0%) R/R patients required pre-emptive treatment with ganciclovir or foscarnet. Half of the ND-AML patients experienced sepsis at a median of 78 days (range, 8 to 277) after allo-HCT, as did 7 out of 40 (17.5%) R/R-AML patients. In addition, pneumonia and BK viruria occurred in five (50%) and three (30%) ND-AML patients, respectively, and three (7.5%) and nine (22.5%) R/R-AML patients, respectively (Table 2). In the ND-AML group, two patients died, one due to infection and the other due to relapse. Of the 14 deaths in the R/R-AML group, 9 were due to relapse, 3 were from infection, 1 was due to pneumonia, and 1 was due to GVHD.

### 3.5. Predicting Factors for Post-Transplant Outcomes

Univariate and multivariate analyses were conducted to evaluate factors that contribute to OS, RFS, CIR, and NRM (Table 3). We first analyzed the whole cohort. The factors associated with a statistically significant impact on OS and RFS in univariate analysis were VEN-HMA cycle (*p* = 0.034) and the response to VEN-HMA therapy before allo-HCT (*p* = 0.001). Although the difference in CIR between the two groups was not statistically significant, it tended to be lower in the group with more than three VEN-HMA cycles compared to the group with less than three VEN-HMA cycles (1-year CIR, 11.8% vs. 35.8%). In multivariate analysis, the factors associated with a statistically significant impact on OS were VEN-HMA cycle (*p* = 0.021, HR = 0.089, 95% CI 0.011 to 0.691), ELN risk group (*p* = 0.041, HR = 0.264, 95% CI 0.073 to 0.947), and the response to VEN-HMA therapy before allo-HCT (*p* = 0.003, HR = 9.745, 95% CI 2.218 to 42.810). In addition, the factors associated with a statistically significant impact on RFS were prior allo-HCT (*p* = 0.045, HR = 2.704, 95% CI 1.023 to 7.142) and the response to VEN-HMA therapy before allo-HCT (*p* = 0.004, HR = 8.627, 95% CI 1.983 to 37.520). The response to VEN-HMA therapy prior to allo-HCT was the only factor associated to CIR (*p* < 0.001, HR = 19.750, 95% CI 3.986 to 97.850), while no significant factor was identified in statistical relation to NRM. Age at HCT, sex, setting, conditioning, and donor type were not significantly associated with any outcome in any patient group.

Given the small number of patients in the ND group, we did not perform analyses and instead focused on the subset of patients with R/R-AML. In the univariate analysis, HCT-CI (*p* = 0.006) and the response to VEN-HMA therapy before allo-HCT (*p* = 0.001) were found to be associated with a statistically significant impact on OS and RFS in the R/R-AML cohort. In addition, prior HCT (*p* = 0.025) and HCT-CI (*p* = 0.025) were found to be associated with a statistically significant impact on NRM. In multivariate analysis, the response to VEN-HMA therapy prior to allo-HCT was associated with RFS (*p* = 0.045, HR = 4.528, 95% CI 1.028 to 19.940) and CIR (*p* = 0.024, HR = 10.420, 95% CI 1.359 to 79.930) in R/R-AML cohort (Appendix A).

## 4. Discussion

Among patients who receive allo-HCT after VEN-HMA treatment, determining the effect of VEN-HMA on post-HCT outcomes in terms of both efficacy and toxicity is an important issue [4,13,14,15,16,32]. Several previous studies have reported allo-HCT outcomes when VEN-HMA was used as a bridge to allo-HCT and Table 4 summarizes the results of those studies. For patients with ND-AML whose median age was 65 to 71.7 years at HCT, the reported 1-year post-HCT outcomes ranged from 63.2% to 76.3% for OS, 58.0% to 73.2% for RFS, and 11.0% to 19.1% for NRM [4,15,16]. In our study, the median age of ND-AML patients who received allo-HCT following VEN-HMA treatment was 67 years at transplant, and the observed 1-year OS, RFS, and NRM after HCT were 77.1%, 80.0%, and 10.0%, respectively, which is similar to the results of previous studies. Pasvolsky O. et al. retrospectively compared post-transplant outcomes when VEN-AZA and IC were used as pretransplant AML treatment modalities. They found that the outcomes between those groups were similar, with 1-year post-HCT OS and RFS of 70.8% and 54%, respectively, for the IC group and 63.2% and 58%, respectively, for the VEN-AZA group. Of note, although the median age of the IC group (58.4 years) was significantly younger than that of the VEN-AZA group (71.7 years), the probability of NRM was comparable between the groups (11.8% for the IC group and 19.1% for the VEN-AZA group) [4]. Collectively, these results indicate that allo-HCT following VEN-HMA therapy is a feasible strategy for elderly ND-AML patients because of its efficacy and acceptable NRM rate.

Few data about R/R-AML patients have been available about post-HCT outcomes after salvage VEN-HMA treatment and no previous study has analyzed just R/R-AML cases. A few studies included R/R-AML patients [13,14], but they analyzed them together with ND-AML patients, which makes it difficult to determine the post-HCT outcomes of R/R-AML patients. For example, Sandhu K.S. et al. analyzed 19 ND- and 13 R/R-AML patients with a median age of 62 years and found CR/CRi rates after VEN-AZA of 68.8% for ND-AML and 31.3% for R/R-AML. After allo-HCT, they reported a 1-year OS of 62.5%, RFS of 43.8%, CIR of 37.5%, and NRM of 18.8% among all their patients [13]. In addition, Kennedy V.E, et al. studied 46 ND-AML and 42 R/R-AML patients, including 5 cases of second transplantation. The median age of their cohort was 67 years, and all received transplants after showing any response (CR, CRi, or MLFS). After allo-HCT, the 1-year OS, CIR, and NRM were reported to be 78%, 36%, and 17%, respectively [14]. In our study, among all patients and R/R-AML patients, the median age at transplantation was 54 and 51 years, respectively, and the proportion of patients achieving any response to VEN-HMA prior to allo-HCT was 92.0% and 92.5%, respectively. In our population, post-transplant outcomes after 1 year were 63.7%, 59.3%, 28.5%, and 12.2% for OS, RFS, CIR, and NRM, respectively, among the entire cohort, and 60.5%, 54.3%, 33.0%, and 12.7%, respectively, among R/R-AML patients.

Compared with the previous two studies, the median age of 54 years in our study was younger, but, considering that 15 people (30%) were receiving a second transplantation and R/R-AML patients accounted for 80% of the whole cohort, our study showed comparable or even superior outcomes. Although we did not analyse the post-HCT outcomes of R/R-AML patients who received IC as a bridge to allo-HCT, we previously reported a retrospective comparison of salvage IC versus VEN combination in patients with R/R-AML, and, in that study, we observed that the percentage of patients who underwent HCT with blast clearance was significantly higher after taking the VEN combination (86.5%) than IC (62.3%) (*p* = 0.010) [10]. Taken together, the results from this study and previous studies suggest that VEN-HMA can provide an effective and safe bridge to allo-HCT for patients with R/R-AML.

In this study, the incidence of both acute and chronic GVHD was higher in patients with R/R-AML than in those with ND-AML, and treatments for GVHD were more often required in R/R-AML patients as well. Given that MAC accounted for 90% and 50% of the conditioning regimens used for R/R- and ND-AML patients, the higher incidence of GVHD observed in the R/R-AML group can be explained in part by the difference in conditioning intensity between the groups [33,34], in addition to the inherent risk of R/R disease itself and second transplantation for GVHD occurrence. Overall, the incidence of GVHD in allo-HCT after VEN-HMA does not seem to differ significantly from that after IC in our institution [28,35]. In other words, the occurrence of GVHD might not be largely influenced by the treatment regimens that patients receive prior to allo-HCT. Instead, it seems to be more affected by traditional risk factors for GVHD occurrence. When we compared our ND- and R/R-AML patients, we found that infectious events occurred more frequently in ND-AML patients than R/R-AML patients. In this study, patients with ND-AML received HCT from HID more often than patients with R/R-AML did (40.0% vs. 20.0%), and it is well known that, compared with other donor types, transplantation from HID can increase the possibility of CMV infection [20,35]. Otherwise, it can be explained that the higher dose of ATG used in HID transplantation, compared with MSD or UD transplantation, in our study could be a cause of the more frequent CMV infections, which is supported by recent data showing a similar frequency of CMV infections in MUD and HID when using the same RIC regimen and ATG dose [36]. Nevertheless, despite concerns about the immunosuppressive effects of venetoclax [37,38], the incidence of CMV infection observed in this study was not higher than that with post-IC transplantation reported in other studies [20,28,35]. As suggested with GVHD complications, traditional risk factors, such as donor type and ATG dosage, are likely to play a greater role in the occurrence of post-transplant infectious complications than the regimens used prior to allo-HCT. Because CMV activation can increase morbidity and hasten mortality, the use of prophylactic agents, such as LTM, should be actively considered in patients at a high risk of infection who are undergoing transplantation [39].

Overall, we observed that allo-HCT following VEN-HMA generally provided good outcomes in both ND- and R/R-AML patients. However, the incidence of post-HCT relapse remained higher among patients with R/R-AML than ND-AML, emphasizing the need for active post-HCT intervention in that group of patients. In fact, to reduce relapses in this high-risk population, various pre-, peri-, and post-transplantation strategies are emerging [40], and triplet treatment using target agents has recently been proposed [41,42]. Furthermore, several investigations of maintenance settings after allo-HCT have been conducted [43,44,45,46]; in particular, the results of the ongoing phase 3 VEN-AZA maintenance therapy study (VIALE-T, NCT04161885) are highly anticipated and should be noted.

We acknowledge that this study has several limitations. When analyzing the response to pre-HCT VEN-HMA, we included only patients who were treated with VEN-HMA as their last treatment, excluding patients who died during their last treatment or did not receive allo-HCT. Therefore, the response rate to VEN-HMA might be overestimated. Moreover, given the nature of a retrospective analysis, our analysis of GVHD and other complications was record-dependent, and data on long-term complications are insufficient due to the short follow-up period. In addition, we did not compare post-HCT outcomes by genetic subgroup and we did not directly compare our results with those from groups receiving treatment other than VEN-HMA before transplantation. Lastly, because of our small sample size, the statistical significance of the factors affecting allo-HCT outcomes is unclear, and the findings from this study cannot be generalized.

## 5. Conclusions

In summary, this study shows that allo-HCT following VEN-HMA could be an effective strategy for patients with ND-AML who are not suitable for IC, as well as for R/R-AML patients. Post-transplant GVHD and infectious complications seemed to be mainly influenced by traditional risk factors rather than AML treatment itself prior to allo-HCT. However, the post-HCT relapse rate was non-negligible among R/R-AML patients, so the results of ongoing investigations targeting those patients at a high risk of relapse should be awaited.

## Figures and Tables

**Figure 1 cancers-15-01666-f001:**
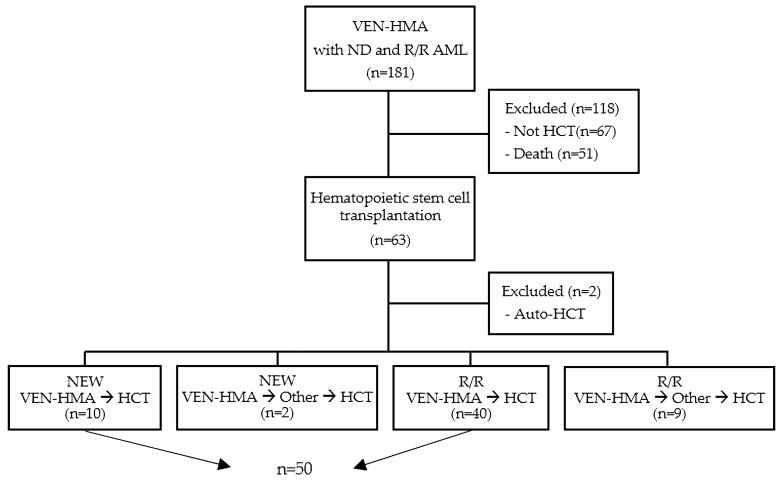
Flow diagram for patient selection. VEN, venetoclax; HMA, hypomethylating agents; ND, newly diagnosed; R/R, relapse/refractory; HCT, hematopoietic stem cell transplantation.

**Figure 2 cancers-15-01666-f002:**
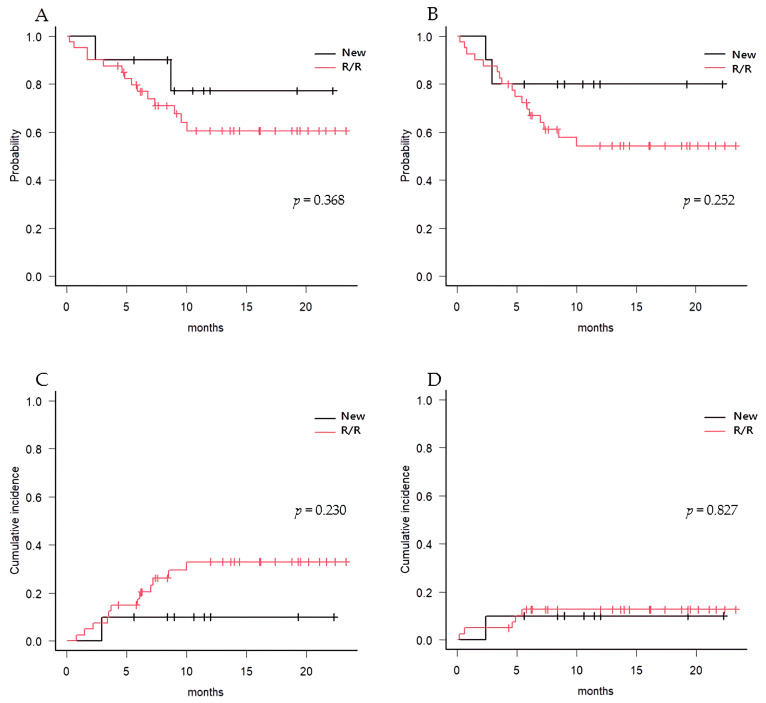
Outcomes of allo-HCT after VEN-HMA. (**A**) Overall survival, (**B**) Relapse-free survival, (**C**) Cumulative incidence of relapse, (**D**) Nonrelapse mortality in newly diagnosed and relapse/refractory AML, respectively.

**Figure 3 cancers-15-01666-f003:**
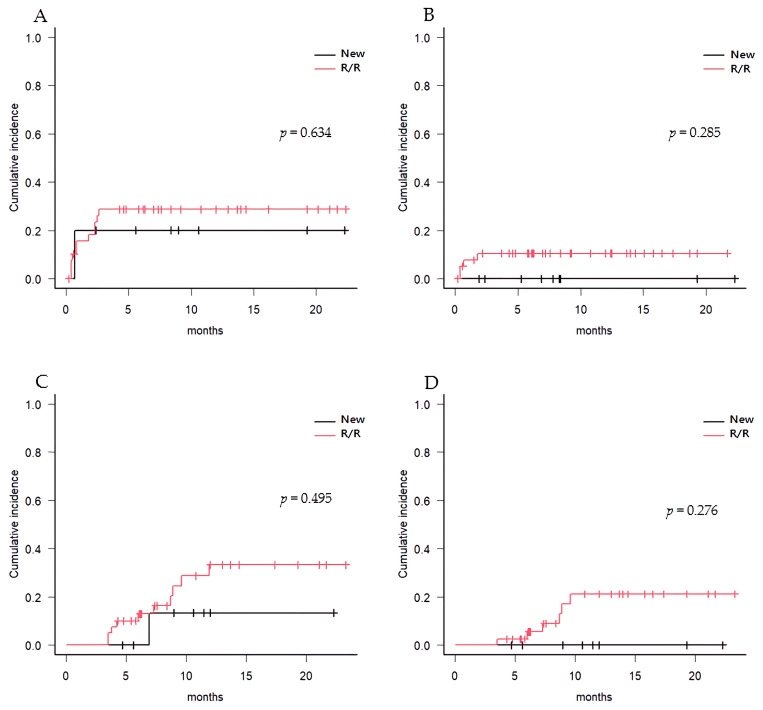
Cumulative incidence of acute GVHD and chronic GVHD. (**A**) aGVHD grade II-IV, (**B**) aGVHD grade III-IV, (**C**) cGVHD grade moderate-severe, (**D**) cGVHD grade severe in newly diagnosed and relapse/refractory AML, respectively.

**Table 1 cancers-15-01666-t001:** Baseline characteristics of patients and allo-HCT.

	All	New	R/R	*p*-Value
Total Number	50	10	40	
Sex				0.308
Male	27 (54.0%)	7 (70.0%)	20 (50.0%)	
Female	23 (46.0%)	3 (30.0%)	20 (50.0%)	
HCT number				0.022
first	35 (70.0%)	10 (100%)	25 (62.5%)	
second	15 (30.0%)	0	15 (37.5%)	
AML type				0.496
De novo	47 (94.0%)	9 (90.0%)	38 (95.0%)	
Secondary	3 (6.0%)	1 (10.0%)	2 (5.0%)	
ELN (2022) risk group				0.848
Favorable	6 (12.0%)	1 (10.0%)	5 (12.5%)	
Intermediate	21 (42.0%)	5 (50.0%)	16 (40.0%)	
Adverse	23 (46.0%)	4 (40.0%)	19 (47.5%)	
Cytogenetic risk group				0.793
Favorable	1 (2.0%)	0	1 (2.5%)	
Intermediate	37 (74.0%)	7 (70.0%)	30 (75.0%)	
Adverse	12 (24.0%)	3 (30.0%)	9 (22.5%)	
Prior chemotherapy line before VEN-HMA 0 1 2	10 (20.0%) 33 (66.0%) 7 (14.0%)	10 (100%) 0 0	0 33 (82.5%) 7 (17.5%)	<0.001
VEN combination				1.000
AZA	1 (2.0%)	0	1 (2.5%)	
DEC	49 (98.0%)	10 (100%)	39 (97.5%)	
VEN-HMA cycles				
Median	3	4	2	<0.001
Range	1–7	3–7	1–6	
Age at HCT				
Median	54	68	51	<0.001
Range	23–73	66–73	23–68	
Response before HCT				0.794
Active disease	4 (8.0%)	1 (10.0%)	3 (7.5%)	
Response disease	46 (92.0%)	9 (90.0%)	37 (92.5%)	
CR/CRi without MRD	21 (42.0%)	4 (40.0%)	17 (42.5%)	
CR/CRi with MRD	11 (22.0%)	5 (50.0%)	6 (15.0%)	
MLFS without MRD	7 (14.0%)	0 (0%)	7 (17.5%)	
MLFS with MRD	7 (14.0%)	0 (0%)	7 (17.5%)	
HCT-CI				0.283
0	17 (34.0%)	3 (30.0%)	14 (35.0%)	
1–2	14 (28.0%)	1 (10.0%)	13 (32.5%)	
≥3	19 (38.0%)	6 (60.0%)	13 (32.5%)	
Donor type				1.000
MSD	9 (18.0%)	1 (10.0%)	8 (20.0%)	
UD *	25 (40.0%)	5 (50.0%)	20 (50.0%)	
HID	12 (24.0%)	4 (40.0%)	8 (20.0%)	
CBT	4 (8.0%)	0	4 (10.0%)	
Conditioning regimen				0.010
MAC	41 (82.0%)	5 (50.0%)	36 (90.0%)	
RIC	9 (18.0%)	5 (50.0%)	4 (10.0%)	
CMV prophylaxis				0.663
Yes	40 (80.0%)	9 (90.0%)	31 (77.5%)	
No	10 (20.0%)	1 (10.0%)	9 (22.5%)	

HCT, hematopoietic stem cell transplantation; AZA, azacitidine; DEC, decitabine; HMA, hypomethylating agents; CR, complete remission; CRi, complete remission with incomplete count recovery; MLFS, morphologic leukemia-free state; MRD, measurable residual disease; HCT-CI, hematopoietic cell transplantation–comorbidity index; MSD, matched sibling donor; HID, haplo-identical donor; CBT, cord blood transplant; MAC, myeloablative conditioning; RIC, reduced-intensity conditioning; CMV, Cytomegalovirus; * UD (unrelated donor) = MMUD (mismatched unrelated donor) + MUD (matched unrelated donor).

**Table 2 cancers-15-01666-t002:** Outcomes of HCT after VEN-HMA combination.

	All	New	R/R	*p*-Value
Total Number	50	10	40	
OS				
Median	NR	NR	NR	0.368
1 year [95% CI]	63.7% [47.3–76.3%]	77.1% [34.5–93.9%]	60.5% [42.0–74.7%]	
RFS				
Median	NR	NR	NR	0.252
1 year [95% CI]	59.3% [43.6–72.1%]	80.0% [40.9–94.6%]	54.3% [36.7–68.9%]	
CIR				
Median	NR	NR	NR	0.230
1 year [95% CI]	28.5% [16.0–42.4%]	10.0% [4.0–37.6%]	33.0% [18.0–48.7%]	
NRM				
Median	NR	NR	NR	0.827
1 year [95% CI]	12.2% [4.9–23.0%]	10.0% [0.5–37.4%]	12.7% [4.6–25.3%]	
Cumulative incidence of aGVHD				
II-IV at 100 days	28.4%	20.0%	30.3%	0.634
III-IV at 100 days	8.4%	0.0%	10.7%	0.285
Cumulative incidence of cGVHD				
Mod-Sev at 1 year	37.4%	15.6%	39.3%	0.495
Severe at 1 year	20.9%	0.0%	23.9%	0.276
Infectious complications				
CMV DNAemia	22 (44.0%)	7 (70.0%)	15 (37.5%)	
CMV Treatment	9 (18.0%)	5 (50.0%)	4 (10.0%)	
Bacteremia	12 (24.0%)	5 (50.0%)	7 (17.5%)	
Pneumonia	8 (16.0%)	5 (50.0%)	3 (7.5%)	
BK viruria	12 (24.0%)	3 (30.0%)	9 (22.5%)	

OS, overall survival; RFS, relapse-free survival; CIR, cumulative incidence of relapse; NRM, nonrelapse mortality; NR, not reached; aGVHD, acute graft-versus-host disease; cGVHD, chronic graft-versus-host disease; Mod, moderate; Sev, severe; CMV, cytomegalovirus.

**Table 3 cancers-15-01666-t003:** Univariate and multivariate analyses for OS, RFS, CIR, and NRM.

	OS			RFS			CIR			NRM		
	Univariate	Multivariate		Univariate	Multivariate		Univariate	Multivariate		Univariate	Multivariate	
	*p*-Value	*p*-Value	HR [95% CI]	*p*-Value	*p*-Value	HR [95% CI]	*p*-Value	*p*-Value	HR [95% CI]	*p*-Value	*p*-Value	HR [95% CI]
Age at HCT	0.894	-	-	0.887	-	-	0.410	-	-	0.440	-	-
Sex	0.472	-	-	0.590	-	-	0.620	-	-	0.850	-	-
Setting (R/R or New)	0.377	-	-	0.265	-	-	0.270	-	-	0.830	-	-
VEN-HMA cycle (>3 or ≤3)	0.034	0.021	0.089 [0.011–0.691]	0.026	0.036	0.196 [0.043–0.903]	0.105	-	-	0.996	-	-
Prior HCT (Yes or no)	0.111	-	-	0.099	0.045	2.704 [1.023–7.142]	0.420	-	-	0.210	-	-
ELN risk group (Poor or others)	0.086	0.041	0.264 [0.073–0.947]	0.154	-	-	0.490	-	-	0.155	-	-
HCT-CI (≥3 or <3)	0.128	0.087	2.724 [0.864–8.587]	0.401	0.517	1.468 [0.460–4.687]	0.700	0.427	0.588 [0.159–2.179]	0.120	0.185	3.267 [0.568–18.810]
Response at HCT (No or response)	0.001	0.003	9.745 [2.218–42.810]	<0.001	0.004	8.627 [1.983–37.520]	0.008	<0.001	19.750 [3.986–97.850]	0.380	0.402	2.610 [0.276–24.650]
Conditioning (RIC or MAC)	0.664	-	-	0.849	-	-	0.910	-	-	0.870	-	-
Donor type (Others or matched)	0.632	-	-	0.182	-	-	0.180	-	-	0.680	-	-

OS, overall survival; RFS, relapse-free survival; CIR, cumulative incidence of relapse; NRM, nonrelapse mortality; VEN, venetoclax; HMA, hypomethylating agents; HCT, hematopoietic stem cell transplantation; R/R, relapse/refractory; ELN, European Leukemia Net; HCT-CI, hematopoietic cell transplantation-comorbidity index; RIC, reduced-intensity conditioning; MAC, myeloablative conditioning.

**Table 4 cancers-15-01666-t004:** Previous studies assessing outcomes of allo-HCT after VEN-HMA.

Authors, Year [Reference No.] (Study Design)	Patients N (ND/RR)	Age at HCT (Range)	Cycle Median N (Range)	Prior HCT N (%)	HCT-CI (%)	Response at HCT	Condition -ing	Donor	Median f/u Duration	OS	RFS	CIR	NRM	CMV N (%)	CI of aGVHD (%)	CI of cGVHD (%)
Pasvolsky, 2022 [4] (Retrospective)	24 (24/0)	71.7 (43–76)	(1–4)	0	≥3 39%	CR 100%	MAC 12% RIC 88%	MSD 21% MUD 67% HID 13%	8.0 months	1 yr 63.2%	1 yr 58%	-	1 yr 19.1%	-	6 months 58%	1 yr 40%
Pollyea, 2022 [15] (Retrospective)	21 (21/0)	65 (60–73)	3 (1–19)	0	≥2 57%	CR/CRi 81% MLFS 19%	MAC 9.5% RIC 57.1% NMA 33.4%	MSD 33% CBT 67%	22.9 months	NR	-	1 yr 20%	1 yr 11%	-	II–IV; 48% III–IV; 10%	Any; 43% Mod-Sev; 10%
Winters, 2022 [16] (Retrospective)	29 (29/0)	65 (22–73) (at Dx.)	3 (1–19)	0	-	CR 86% MLFS 14%	MAC 31.0% RIC 41.4% NMA 27.6%	MSD 28% HID 3% CBT 69%	14.3 months	1 yr 76.3%	1 yr 73.2%	-	-	-	-	-
Current study (Retrospective)	50 (10/40)	54	3 (1–7)	15 (30%)	≥3 38%	CR/CRi 64% MLFS 28% R/R 8%	MAC 82% RIC 18%	MSD 18% UD 40% HID 24% CBT 8%	13.7 months	1 yr 63.7%	1 yr 59.3%	1 yr 28.5%	1 yr 12.2%	9 (18.0%) treatment	100 days II-IV; 28.4% III-IV; 8.4%	1 yr Any; 55.2% Mod-Sev; 37.4%
Sandhu, 2020 [13] (Retrospective)	32 (19/13)	62 (18–73)	2 (1–12)	0	≥2 31.3%	CR/CRi 68.8% Refractory 31.3%	MAC 6.2% RIC 62.5% NMA 31.3%	MSD 25% MUD 68.8% HID 6.3%	14.4 months	1 yr 62.5%	1 yr 43.8%	1 yr 37.5%	1 yr 18.8%	5 (15.6%) reactivation	100 days II-IV; 43.8% III-IV; 21.9%	1 yr Any; 31.3% Ext.; 28.1%
Kennedy, 2022 [14] (Retrospective)	88 (46/42)	67 (24–77)	3 (1–13)	5 (6%)	≥3 35%	CR 70% CRi 15% MLFS 15%	MAC 28% RIC 55% NMA 17%	MSD 25% MUD 55% HID 19% CBT 1%	10.9 months	1 yr 73%	-	1 yr 36%	1 yr 17%	-	-	-

ND, newly diagnosed; RR, relapse/refractory; HCT, hematopoietic stem cell transplantation; N, number; HCT-CI, hematopoietic cell transplantation–comorbidity index; f/u, follow-up; Dx., diagnosis; OS, overall survival; RFS, relapse-free survival; CIR, cumulative incidence of relapse; NRM, nonrelapse mortality; CMV, Cytomegalovirus; CI, cumulative incidence; aGVHD, acute graft-versus-host disease; cGVHD, chronic graft-versus-host disease; CR, complete remission; CRi, complete remission with incomplete count recovery; MLFS, morphologic leukemia-free state; MAC, myeloablative conditioning; RIC, reduced-intensity conditioning; NMA, non-myeloablative conditioning; MSD, matched sibling donor; MUD, matched unrelated donor; UD, unrelated donor; HID, haplo-identical donor; CBT, cord blood transplant; Ext., extensive; Mod, moderate; Sev, severe.

## Data Availability

The data presented in this study are available on request from the corresponding author. The data are not publicly available owing to privacy and ethical reasons.

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
