# Peer review of "A Successful Bridge Therapy Combining Hypomethylating Agents with Venetoclax for Adult Patients with Newly Diagnosed or Relapsed/Refractory Acute Myeloid Leukemia"

_cancers, 2023, doi:10.3390/cancers15061666_

Round 1

Reviewer 1 Report

The authors report a retrospective study of 50 patients treated with VEN/HMA followed by allogenic transplant. The study in interesting for the hematologists in charge of AML patients.

Major comments:

Regarding the 10 patients treated upfront with VEN/AZA. What were the reasons/comorbidities for not giving them intensive chemotherapy while they were considered eligible for allo transplant ? This should be discussed.

What was number of line of therapies for R/R AML?

Do the authors have molecular (NGS) data for a subset of patients or for the whole patients set?

What about the impact of cytogenetics or molecular data on survival in univariate and multivariate analyses? Why not using ELN 2022 which include molecular data instead of ELN 2017?

Is the number of VEN/HMA cycles having an impact on duration of response after HCT?

Authors mentioned MRD in the M&M section? Do they have data on MRD to display in the results?

Minor comments: 

I would simplify the paragraph 3.5 by rephrasing it as suggested below : 

We first analyzed the whole cohort. The factors associated with a statistically significant impact on OS, RFS, CIR, and NRM in univariate analysis were X, X, X… The factors associated with a statistically significant impact on OS, RFS, CIR, and NRM In multivariate analysis, were X, X, X…

Given the small number of patients in the ND group, we did not performed analyzes and we  focus on R/R AML subset of patients. In univariate analysis the factors associated with a statistically significant impact on OS, RFS, CIR, and NRM in the whole cohort were X, X, X… In multivariate analysis, the factors associated with a statistically significant impact on OS, RFS, CIR, and NRM in the whole cohort were X, X, X…

Replace "bloodstream infection" by "sepsis" (l 233)

"In the univariate analysis, the response to VEN-HMA therapy before allo-HCT was independently" (l. 247). Remove "independently" as this is univariate analysis

"The total number of allo-HCT was significantly associated with OS" (l. 32 and l. 249). Replace "total number of allo" by "prior allo-HCT"

Reviewer 2 Report

This is an interesting piece of descriptive evidence on the role of VEN+decitabine treatment as bridge therapy to HCT in AML patients, either newly-diagnosed or in the realpse setting. It gives a clear view of the potential usefulness of this combination in clinical practice, as a tool to drive the patients to HCT without using intensive CT. Their results are very encouraging.
As discussed by the authors, the main limitation of this work is the small sample size considered, specially the low number of newly diagnosed patients analyzed, that implies very wide confidence intervals in the estimation of their response proportions. In addition, the lack of a control group precludes any inference on its relative value as compared to the more traditional approach (ICT), but fortunately some randomized clinical trials are on the way to address this issue.
Nevertheless, the information included in the manuscript is of great interest as it describes very precisely the clinical outcomes of their patients. In addition, the authors have made an effort to analyze the main covariates linked to such clinical evolution.
The manuscript is well written and easy to read. I just would point out the inconsistent use of SCT (“Response to SCT”) instead of HCT in the Supplementary Table 1 (S1).

Round 2

Reviewer 1 Report

The authors made significant changes to improve their manuscipt. I don't have further comments.